# Evaluating healthcare professionals' readiness for community-based kangaroo mother care in resource-limited setting

Hagos T. Atalay[1,2¤b] *, Victoria J. Kain[1¤a], Amanda G. Carter[1¤a]

1 School of Nursing and Midwifery, Griffith University, Brisbane, Queensland, Australia, 2 School of Nursing, Aksum University, Axum, Tigray, Ethiopia

☺ These authors contributed equally to this work.
¤a Current Address: School of Nursing and Midwifery, Griffith University, Brisbane, Queensland, Australia
¤b Current Address: School of Nursing, Aksum University, Axum, Tigray, Ethiopia
* hagostasew.atalay@griffithuni.edu.au

## Abstract

### Background

Neonatal mortality remains a major public health challenge in low-resource settings. Community-Based Kangaroo Mother Care (cKMC) improves survival outcomes for preterm and low-birthweight infants. However, its implementation is hindered by healthcare professionals' readiness and the lack of validated tools to assess and support cKMC. This study primarily aimed to assess the readiness of healthcare professionals for the implementation of cKMC. In addition, this study developed and performed preliminary validation of an instrument to measure healthcare professionals' capability, perception, knowledge, and engagement with cKMC.

### Methods

A cross-sectional study was conducted among 593 healthcare professionals in northern Ethiopia. A 30-item instrument was developed through literature review, expert consultation, and cognitive interviews. Preliminary psychometric evaluation included content validity, internal consistency reliability, and construct validity testing. Sampling adequacy was confirmed using the Kaiser–Meyer–Olkin measure of 0.82 and Bartlett's test of sphericity ($P<0.001$). Descriptive analyses, Kruskal–Wallis tests, and generalised linear models were applied to examine subscales of readiness and their predictors.

### Results

The survey instrument assessing healthcare professionals' capability, perception, knowledge, and engagement regarding cKMC demonstrated strong psychometric performance (CVI = 0.97; Cronbach's α = 0.85). An evaluation of construct validity

**Data availability statement:** All relevant data are within the paper and its Supporting Information files.

**Funding:** The author(s) received no specific funding for this work.

**Competing interests:** The authors have declared that no competing interests exist.

through factor analysis generated four factors: capability, perception, knowledge, and engagement. Total and subscale scores correlated significantly. Only 13.2% of participants had read the national the cKMC guideline, and 73.7% had not received formal training; 55.6% reported inadequate workplace support. In multivariable analyses, prior cKMC experience predicted higher perception scores (p = .029); age predicted knowledge (p < .001); and both age and work experience were associated with higher engagement (p < .001).

## Conclusion

The instrument demonstrated satisfactory preliminary reliability and construct validity, providing a context-appropriate measure of healthcare professionals' readiness to implement cKMC. Although perceptions were positive, substantial gaps in knowledge, familiarity with guidelines, and practical capabilities highlight the need for structured training and institutional support to enhance community-based neonatal care.

## Introduction

Neonatal mortality continues to pose a significant global health challenge, particularly in low-resource settings where access to quality maternal and newborn care is limited [1,2]. In Ethiopia, the neonatal mortality rate of 33 per 1,000 live births remains markedly higher than the 2–3 per 1,000 reported in high-income countries [3–6]. The leading contributors to neonatal deaths are preterm birth and low birthweight [7].

Kangaroo Mother Care (KMC) is an approach for caring of premature and low birthweight neonates that involves continuous skin-to-skin contact for at least 8 hours a day, coupled with exclusive breastfeeding [8]. This method has emerged as a cost-effective, evidence-based intervention to improve outcomes of premature and low birthweight neonates, endorsed by the World Health Organisation (WHO) [8,9]. Adopting the core principles of KMC in the community or at home is referred to as community-based kangaroo mother care (cKMC) [8], which has the potential to reduce neonatal mortality by 30% [10,11], decrease hypothermia and neonatal infections, and improve rates of exclusive breastfeeding [12], and a reduction in postpartum depression among mothers [13].

Although facility births are recommended, many mothers in low- and middle-income countries continue to give birth at home, highlighting the need for community-based interventions such as cKMC [14]. Despite cKMC being also endorsed by WHO to support mothers in giving birth at home and continuing care after leaving health facilities [8] and its proven effectiveness, the implementation of cKMC remains limited [15,16]. Existing literature has largely focused on facility-based KMC, leaving a gap in understanding the scalability and sustainability of cKMC in real-world, low-resource settings [15].

The role of healthcare professionals in facilitating cKMC is not well defined. Healthcare professionals play an essential part in educating families, demonstrating techniques, providing hands-on support, and advocating for the integration of policies

and resources [17–19]. Despite their significance as stakeholders in the implementation process, there is currently no reliable instrument to assess health professionals' capability, perception, knowledge, and engagement needs in the community and/or home environments, existing tools predominantly focus on facility-based contexts [17–19].

The health system in Tigray, Ethiopia is severely under-resourced, with challenges compounded by conflict-related destruction of infrastructure. Many health facilities have been damaged, looted, or vandalised, and healthcare professionals continue to operate under extreme conditions with limited support [20]. Therefore, addressing the implementation gaps of cKMC is essential to inform the development of targeted training programs, policy frameworks, and interventions that support broader and more effective implementation of cKMC.

This study primarily aimed to assess the readiness of healthcare professionals for the implementation of cKMC. In addition, this study developed and performed preliminary validation of an instrument to measure healthcare professionals' capability, perception, knowledge, and engagement with cKMC, providing a foundation for targeted capacity-building and evidence-based strategies to improve neonatal outcomes.

## Materials and methods

### Study design

A cross-sectional study was employed using the consensus-based checklist for reporting survey studies (CROSS) guidelines to ensure transparency, rigour, and reproducibility [21]. This design was chosen to assess healthcare professionals' readiness and conduct preliminary validation of the healthcare professionals' instrument on capability, perception, knowledge, and engagement regarding cKMC at a single time point.

### Study settings and population

The study was conducted in communities, health posts, health centres, and primary hospitals in Central and North-western Tigray, Ethiopia, where maternal and child health programs are actively implemented. In community settings, KMC implementation involves health extension workers (HEWs), supported by midwives, nurses, health officers, and physicians engaged in primary health care. Health extension workers are frontline health professionals responsible for delivering community-based services through the health extension program in Ethiopia [22]. Their role includes home visits, counselling mothers and families on skin-to-skin care, early identification of low-birth-weight infants, follow-up after facility discharge, and referral to health facilities when complications arise. Midwives and nurses provide technical guidance and continuity of care during antenatal, postnatal, and outreach services, while health officers and physicians support supervision, mentoring, and integration of cKMC within the broader maternal and newborn care system.

Participants were selected using a stratified sampling method. It is a method where a population is divided into distinct subgroups (strata) based on shared characteristics, and a random sample is taken from each subgroup [23]. A stratified sampling approach was used to ensure appropriate representation of healthcare cadres involved in community and primary health care. Participants were first stratified into two groups: health extension workers and other healthcare professionals (midwives, nurses, health officers, and physicians). This stratification was based on differences in roles and workforce distribution and was proportionally allocated to each stratum, and participants were randomly selected from each stratum. The first stratum comprised 257 health extension workers, whereas the second stratum comprised 336 healthcare professionals (midwives, nurses, health officers, and physicians).

Health extension workers constitute the largest cadre in the study areas and are primarily responsible for community-based maternal and newborn health services The second stratum represented other health professionals involved in guidance during facility-based care and during discharge education and support, linking with health extension programs, referral, supervision, and technical support.

Simple random sampling was used to minimise potential sampling bias, which was conducted independently within each stratum. This approach ensured that both community-based and facility-based perspectives were adequately captured, while preventing over- or underrepresentation of any single healthcare group. The stratification strategy was therefore intended to improve representativeness rather than favour a particular group. Professionals working in general or specialised hospitals were excluded to maintain a focus on community and primary care.

## Ethical considerations

The study was conducted in accordance with the ethical principles outlined in the Declaration of Helsinki [24]. Ethical approval was obtained from the Human Research Ethics Committee (HREC) at Aksum University, under IRB number 005/2025, Ethiopia, as well as from the HREC at Griffith University, Australia, with reference number GU=2025/011. A permission letter was secured from the Tigray Health Research Institute and distributed to each health district and participating health facility.

Participants were provided with an information sheet that detailed the study's aims, procedures, duration, and the voluntary nature of participation. Data were stored securely. Contact information for the research team and the ethics committees was also included for any questions or concerns. Participants understood that completing the online survey questionnaire constituted consent.

## Recruitment and data collection

Healthcare professionals were recruited through direct engagement by two trained research assistants, who provided comprehensive explanations regarding the study's purpose and methodology. Participants were given the opportunity to select their preferred mode of participation, which included online surveys, paper-based questionnaires, or interviewer-administered formats. Following their selection, participants were formally enrolled in the study.

Data collection primarily utilised the Qualtrics [25] platform for online surveys, available in both English and Tigrinya, with additional support from paper-based surveys and interviews. In the interviewer-administered format, research assistants read questionnaire items aloud and recorded participant responses directly into Qualtrics. This diverse methodology was designed to ensure inclusivity, accommodating varying levels of literacy and access to technology. The research assistants, possessing experience in conducting data collection across multiple formats, played a critical role throughout this process. Prior to the onset of data collection, the lead researcher (HA) briefed the assistants on the study's objectives, their responsibilities, and the protocols in place to address ethical considerations.

Data were collected from March 13, 2025, to June 15, 2025. The focus of the study was on the development and validation of instruments aimed at assessing the capabilities, perceptions, knowledge, and engagement of healthcare professionals regarding culturally competent Kangaroo Mother Care (cKMC). The resultant instrument, designated as the HiPECK-cKMC, underwent rigorous development and evaluation to ascertain its reliability and validity, comprising 30 items distributed across four dimensions.

## Instrument development and validation process

### Theoretical foundation

The development process was guided by DeVellis's stepped approach to scale development [26] and the Theoretical Domains Framework (TDF) [27]. The TDF includes 14 domains and 128 constructs, which were grouped into four key areas for the purpose of item generation: Knowledge and Role, Attitudes and Beliefs, Skills, and Barriers. These domains ensured comprehensive representation of the behavioural aspects relevant to cKMC implementation.

### Step I: Identifying and defining domains

The initial step involved identifying and defining the domains to be measured. The TDF was selected to maintain conceptual clarity and focus on the intended constructs.

## Step II: Generating the item pool

A deductive approach was used to generate items, informed by a previously conducted systematic literature review and adaptation of existing KMC survey instruments [17–19]. Item adaptation was guided by three key healthcare professionals' focused questionnaires: the survey of kangaroo care practice, knowledge, barriers, and perceptions developed by Engler et al. [18], a healthcare professionals' KMC knowledge and practice survey applied in Indonesia by Utami and Huang [19], and a NICU-based KMC knowledge, practice, and perception survey among nurses by Al-Shehri and Binmanee [17]. These instruments were selected due to their comprehensive assessment of provider knowledge, attitudes, perceived barriers, and reported KMC practices and application across diverse health-care settings.

From these sources, 20 items were adapted that were conceptually relevant to cKMC delivery and could be meaningfully reframed for non-hospital contexts. An additional 25 items were newly generated using domains and constructs from the Theoretical Domains Framework (TDF) [28], informed by evidence from the systematic review to ensure inclusion of behavioural, organisational, and contextual determinants of cKMC implementation. Items were excluded from the facility-based instruments that related to advanced or hospital-level care, including neonatal intensive care unit (NICU) practices. Items were excluded if they (1) explicitly assumed hospital infrastructure, (2) reflected specialist neonatal clinical procedures, or (3) were not feasible within community or primary-care settings.

## Step III: Determining the measurement format

A Likert scale was chosen for its effectiveness in capturing behaviours, attitudes, and perceptions. Each item was presented as a declarative statement with five response options: Strongly Disagree (1), Disagree (2), Unsure/Uncertain (3), Agree (4), and Strongly Agree (5). The neutral midpoint ('Neither agree nor disagree') was intentionally excluded to discourage indirect responses and encourage directional responses, while the 'Unsure/Uncertain' option was retained to differentiate non-attitudes or lack of knowledge from genuine neutrality [29].

## Step IV: Content validity

Content validity was established through an expert panel review who independently assessed item relevance using a structured checklist. A panel of seven experts, all possessing extensive clinical neonatal expertise and experience in instrument development (including five with doctoral qualifications and two doctoral candidates), was chosen for their knowledge in maternal-child health and item construction. This diverse group included four experts from Australia and three from Ethiopia. The expert review panel were provided with a list of draft items, and the content validity index was calculated from their individual ratings, and items were revised accordingly to improve clarity and relevance. Six of the experts provided both quantitative and qualitative feedback, while one contributed only qualitative insight.

The items were assessed for relevance using a 4-point scale, where ratings of 3 or 4 were deemed relevant. A Content Validity Index (CVI) threshold of 0.80 was established [30,31], and the average CVI for the scale was computed. Additionally, the experts offered qualitative feedback regarding clarity, relevance, and cultural appropriateness, which was instrumental in guiding the refinement of the items.

## Step V: Cognitive interviewing

Cognitive interviews were conducted with eight healthcare professionals, who were selected randomly among study participants, to understand their interpretations of the draft survey item [26]. They shared their opinions and interpretations on each item, described their response process, and suggested improvements for clarity and cultural relevance, which informed revisions to simplify medical terminology and enhance cultural appropriateness.

### Step VI: Administering items to the sample

The final instrument was completed by 593 participants, exceeding the recommended ratio of 10 respondents per item for psychometric testing and descriptive analyses [32,33].

### Step VII: Item evaluation reliability and validity

Reliability Testing: Cronbach's alpha was used to assess internal consistency, with coefficients ≥ 0.70 considered acceptable indicators of internal consistency [34,35].

Construct Validity: Initial construct validity was examined through inter-item correlations and exploratory factor analysis (EFA) using Principal Component Analysis (PCA) with Varimax rotation. The rationale for employing PCA in this investigation is rooted in the objectives of exploring a novel instrument and condensing the dataset into more manageable dimensions. This analytical approach facilitates the identification of underlying structures within the data, thereby enhancing interpretability and reducing complexity. The final instrument developed from this process, with 30 items, was named the Healthcare professionals' Instrument for Perception, Engagement, Capability, and Knowledge for Community-based Kangaroo Mother Care (HiPECK-cKMC).

### Internal structure (Inter-Factor Correlations)

The internal structure of the instrument was analysed by evaluating the correlations among subscale scores using the Pearson product-moment correlation coefficient, offering preliminary support for construct validity. Significant positive correlations ($p < 0.05$) were interpreted as indicative of coherence among theoretically related domains, thus providing initial support for construct validity.

### Statistical analysis

Data analysis was conducted using IBM SPSS Statistics software for Windows, version 30.0, to evaluate the instrument and measure healthcare professionals' capabilities, perceptions, knowledge, and engagement. Data were non-normally distributed as assessed by Kolmogorov–Smirnov and Shapiro–Wilk tests, $p < .001$ [36]. Data transformation has been attempted using log and cube root transformations, which did not yield any improvement in normality. The method of analysis used was the Kruskal-Wallis test and the generalised linear model.

Descriptive statistics, including frequencies, percentages, means, and standard deviations, were used to present socio-demographic and participant characteristics. Composite level scores were computed by summing the composite items for each subscale. The Kruskal-Wallis test was conducted to compare rank means across different components associated with socioeconomic and professional characteristics. Post-hoc analyses, Dunn's test was performed, Bonferroni correction was applied, and effect sizes measured by epsilon square ($\varepsilon^2$) were calculated to indicate the magnitude of the observed differences [37,38]. The effect size was classified as negligible (0.00 to 0.01), weak (0.01 to 0.04), moderate (0.04 to 0.16), relatively strong (0.16 to 0.36), strong (0.36 to 0.64), and very strong (0.64 to 1.00) [39].

Generalised linear model was used for regression analyses, which were executed to identify factors predicting healthcare professionals' capability, perception, knowledge, and engagement. Assumptions related to multicollinearity among independent variables were assessed using the variance inflation factor (VIF) and the pairwise Pearson correlation matrix (r). A VIF greater than 10 and a correlation coefficient greater than 0.80 were used as thresholds to indicate multicollinearity [40]. The assumptions were satisfied with VIF values below 5 and correlation coefficients of less than 0.77.

The model's fit and explanatory power were evaluated using the Generalised Linear Model framework with a Gamma distribution and log link. Model fit was assessed using the Pearson Chi-square statistics and deviance measures. Additionally, model performance was evaluated using information criteria such as the Akaike Information Criterion (AIC) and Bayesian Information Criterion (BIC) [41,42].

### Inclusivity in global research

Additional information regarding the ethical, cultural, and scientific considerations specific to inclusivity in global research is included in the Supporting Information (S1 File).

## Results

### Descriptive characteristics of healthcare professionals

A total of 650 healthcare professionals were approached for participation, and 593 completed the survey, yielding a response rate of 91.2%. The median age of participants was 33 years (interquartile range (IQR) = 11), and over half (59.5%) worked in urban health facilities. The majority were health extension workers (43.3%), and 43.2% of healthcare professionals held diplomas, and 37.6% bachelor's degrees. The median duration of professional experience was 10 years (IQR = 12). Most respondents had heard about KMC (92.6%) and cKMC (79.4%). Facility-based KMC experience was reported by 64.9%, and 59.0% reported experience supporting families with cKMC. However, only 13.2% were aware of technical or implementation guidelines for KMC (Table 1).

In terms of training, 73.7% had never received cKMC training. Institutional support was also limited, with 2.9% receiving assistance from their employer and 1.5% from NGOs. Engagement in cKMC activities varied: 46.9% conducted home visits to support families, 31.4% facilitated cKMC during neonatal transport, 51.1% led awareness campaigns, and 41.3% promoted cKMC during antenatal care visits. Participation in conferences for pregnant women was reported by 32.5%, whereas collaboration with village health leaders (7.1%) and health development armies (12.0%) (Table 1).

### Overview of instrument development

The HiPECK-cKMC instrument was developed by mapping 45 draft items to TDF domains and constructs. After reviewing and merging similar items, 34 items were assessed through expert review. The CVI results revealed two items below the 0.80 threshold; one was removed, and one was revised and retained. The mean scale-level CVI was 0.97. Experts also provided qualitative feedback for item clarity. Survey responses from healthcare professionals were collected and analysed using exploratory factor analysis, resulting in the removal of three items with low factor loadings. The final HiPECK-cKMC instrument comprised of 30 items.

### Findings of construct validity

The scale demonstrated strong sampling adequacy for factor analysis, with a KMO value of 0.82, exceeding the 0.6 threshold [43]. Bartlett's Test of Sphericity was significant ($\chi^2$ (496) = 4672.918, p < .001), indicating adequacy of correlations among items for factor analysis [44]. For EFA, cases with missing data on any of the 30 items were excluded via listwise deletion.

Exploratory factor analysis (PCA extraction, Varimax rotation) was conducted to examine the structure of a 30-item instrument assessing healthcare professionals' capability, perception, knowledge, and engagement related to cKMC.

Although the Kaiser criterion suggested up to 10 components (61.2% variance), parallel analysis supported a four-component solution, which this study retained the four component because parallel analysis is considered the most accurate, objective, and robust method for determining the number of factors or components to retain in EFA over eigenvalues and scree plot (Fig 1) [45]. This analysis was conducted using the syntax for SPSS developed by O'Connor (2002) [46]. The four components had eigenvalues exceeding 1, explaining 18.97%, 7.81%, 7.02%, and 5.71% of the variance, respectively (Table 2). The four factors were named according to underlying constructs: Capability, Perception, Knowledge, and Engagement.

**Table 1. Socio-demographic status and professional information about cKMC of the study participants (n = 593).**

| Variable | Categories | n (%) |
|---|---|---|
| Age in years | <21 | 2 (.3) |
| | 21-30 | 228 (38.4) |
| | 31- 40 | 223 (37.6) |
| | 41-50 | 76 (12.8) |
| | >50 | 64 (10.8) |
| Residence | Urban | 353 (59.5) |
| | Rural | 240 (40.5) |
| Profession | Nurse | 112 (18.9) |
| | Midwifery | 184 (31.0) |
| | Health Extension Worker | 257 (43.3) |
| | Health officer | 37 (6.2) |
| | Physician | 3 (0.5) |
| Qualification | Certificate | 107 (18) |
| | Diploma | 256 (43.2) |
| | Bachelor | 223 (37.6) |
| | Master | 6 (1.0) |
| | Doctoral | 1 (0.2) |
| Work experience in years | <5 years | 117 (19.7) |
| | 6-10 years | 204 (34.4) |
| | 11-15 years | 91(15.3) |
| | >15 years | 181 (30.5) |
| Have you heard about KMC? | Yes | 549 (92.6) |
| | No | 44 (7.4) |
| Have you heard about cKMC? | Yes | 471 (79.4) |
| | No | 122 (20.6) |
| Have you supported families about cKMC? | Yes | 350 (59.0) |
| | No | 243 (41.0) |
| Have you seen Ethiopia's "KMC Technical and Implementation Guideline" released in January 2023? | Yes | 78 (13.2) |
| | No | 515 (86.8) |
| How did you obtain the "KMC Technical and Implementation Guideline"? | Email | 7 (1.2) |
| | Telegram | 3 (0.5) |
| | WhatsApp | 12 (2.0) |
| | Training | 50 (8.4) |
| | Colleague | 6 (1.0) |
| | Employer | 17 (2.9) |
| | NGO | 9 (1.5) |
| Have you experienced KMC in a healthcare setting? | Yes | 385 (64.9) |
| | No | 208 (35.1) |
| Have you been trained on cKMC? | Never | 437 (73.7) |
| | One time | 140 (23.6) |
| | Two or more | 16 (2.7) |
| Do you visit families at home to support care for preterm and LBW babies? | Yes | 385 (64.9) |
| | No | 208 (35.1) |
| Do you use cKMC for transporting preterm and LBW neonates? | Yes | 186 (31.4) |
| | No | 407 (68.6) |

*(Continued)*

**Table 1.** (Continued)

| Variable | Categories | n (%) |
|---|---|---|
| Where did you use cKMC for preterm and LBW transport? | Home to a health facility | 48 (8.1) |
| | Home to community (church, market) | 12 (2.0) |
| | Health facility to home | 89 (15.0) |
| | Health facility to health facility | 107 (18.0) |
| | All listed above | 18 (3.0) |
| Have you raised awareness about cKMC? | Yes | 303 (51.1) |
| | No | 290 (48.9) |
| Where did you raise awareness about cKMC? | Antenatal Care (ANC) Visit | 243 (43.3) |
| | Pregnant women conferences | 193 (32.5) |
| | Village health leaders | 42 (7.5) |
| | Health development army | 71 (12.0) |
| | Postnatal care visit | 7 (1.2) |

Certificate is the entry-level qualification for rural health extension workers. It requires completion of 10th grade, one year of formal pre-service training, and passing the Level III Certificate of Competency test (COC).

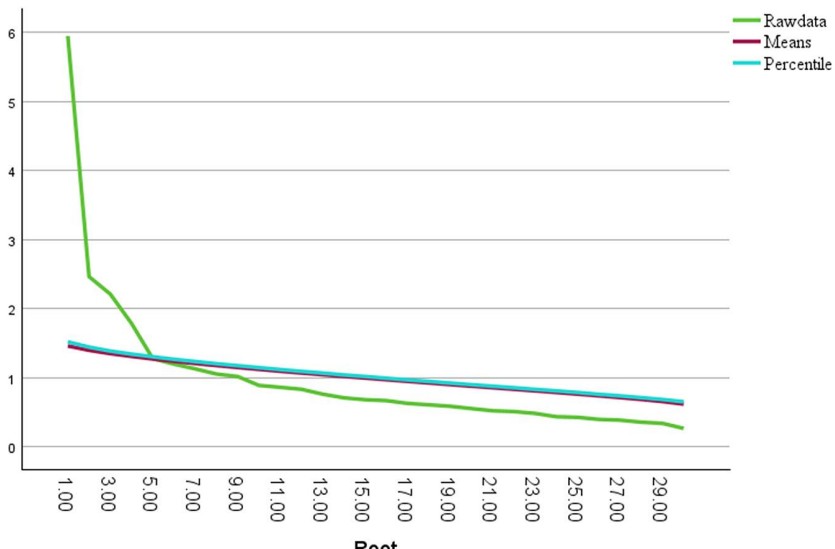

**Fig 1. Parallel analysis scree plot for the HiPECK-cKMC implementation readiness instrument.**

The line graph illustrates a parallel analysis of eigenvalues obtained from actual data alongside those generated randomly using means and percentiles for principal component analysis related to the HiPECK-cKMC instrument. The analysis identifies four components with eigenvalues that surpass both the means and percentiles.

### Internal reliability and item analysis

A reliability analysis was conducted to examine the internal consistency of the 30-item scale assessing healthcare professionals' capability perception, knowledge, and engagement related to cKMC. The overall internal consistency of the scale

was acceptable, with Cronbach's alpha = 0.85, indicating good internal consistency. Cronbach's subscale alphas ranged from 0.67 to 0.80. Cross-loadings were assessed by comparing primary and secondary factor loadings; items were retained if their primary loading was ≥ 0.40 and exceeded all secondary loadings by at least 0.20 [47](Table 2).

## Dimensions of healthcare professionals' readiness on the implementation of cKMC

### Factor 1: Capability

This nine-item factor exhibited strong internal reliability, with a Cronbach's alpha of .78. The total score for this factor was median (IQR) = 31 (8) out of 9–45. The mean item score for this factor was 3.35. This subscale had the lowest mean score among factors, suggesting comparatively lower ratings on capability-related items. Additionally, the correlation between this subscale and the overall scale was substantial, with a correlation coefficient of $r = .77$ ($p < .001$). Only 23.6% of participants reported having received formal training in cKMC, with the lowest scoring mean of 2.39 (1.15). Furthermore, 55.6% of respondents perceived their workplace lacking the necessary resources and support to assist families with cKMC, and 36.8% did not feel competent in supporting families with practice.

### Factor 2: Perception

This eight-item subscale showed strong internal reliability, with a Cronbach's alpha of .77. The total score for this factor was median (IQR) = 28 (7.5) out of a possible 8–40, indicating an item mean of 3.53. Additionally, a positive correlation was observed between this subscale and the overall scale ($r = .72$, $p < .001$). A considerable proportion of respondents (66.5%) perceived cKMC as time-consuming. Additionally, 48.6% believed there are specific medical conditions where cKMC may not be suitable. The lowest mean (SD) was related to community support for families, 3.35 (1.19) and the highest to maternal comfort and privacy, 3.88 (0.92).

### Factor 3: Knowledge

This ten-item factor demonstrated good internal reliability, with a Cronbach's alpha of .80. The overall score for the factor was median (IQR) = 40 (7) of a maximum of 50. This factor had a mean score of 3.91. A positive correlation was found between this subscale related to participants' knowledge of KMC and the total scale, with a correlation coefficient of $r = .72$ ($p < .001$). Only 65.3% of respondents were aware of the recommended minimum duration of cKMC as eight hours per day. Awareness of specially designed baby wraps for safe skin-to-skin care was also limited, with 24% unfamiliar. Furthermore, 20.2% of participants lacked awareness of the core components of cKMC, including skin-to-skin contact and exclusive breastfeeding. The highest mean (SD) for recognising cKMC as safe and effective and promoting breastfeeding, 4.16 (0.82), with lower scores for awareness of the recommended duration, 3.70 (1.00).

Table 2. Summary of item analysis and factor analysis.

| Factor | Item analysis | | | | Construct Validity (factor analysis) | | | Internal reliability: Cronbach's Alpha |
|---|---|---|---|---|---|---|---|---|
| | # of Items | Corrected item total correlation | Corrected item sub-scale correlation range | Subscale–total correlation (Pearson r) | Eigen value | % explained variance | Loading range | |
| 1) Capability | 9 | 0.30-0.55 | 0.47-0.67 | 0.77 | 6.07 | 18.97 | 0.37-0.66 | 0.78 |
| 2) Perception | 8 | 0.30-0.62 | 0.49-0.73 | 0.72 | 2.50 | 7.81 | 0.43-0.75 | 0.77 |
| 3) Knowledge | 10 | 0.38-0.61 | 0.51-0.71 | 0.72 | 2.24 | 7.02 | 0.41-0.65 | 0.80 |
| 4) Engagement | 3 | 0.37-0.52 | 0.72-0.80 | 0.44 | 1.82 | 5.71 | 0.57-0.68 | 0.67 |

## Factor 4: Engagement

This three-item subscale revealed lower internal reliability, with a Cronbach's alpha of .67. This may be due to the small number of items in this factor. The factor's total score was median (IQR) = 12 (4), out of a possible range of 3–15, while the mean for individual items was 4.16. Despite the highest mean item score, internal consistency was lower (alpha = 0.67). Furthermore, a positive correlation was found between this subscale and the total scale, with a correlation coefficient of r = .44 (p < .001). The Engagement subscale demonstrated the highest mean (SD) for the item related to the need for increased financial support from governments and non-governmental organisations, 4.24 (0.79) (Table 3).

## Comparisons, correlation and regression analysis

Between-group comparisons and regression results are summarised in Table 4. The Kruskal–Wallis test examined raw group differences. Generalised linear model (GLM) was conducted using a gamma distribution and a log link function to examine predictors of healthcare professionals' capability, perception, knowledge, and engagement regarding cKMC. The predictors entered into each model included age category, work experience, qualification, residence, training, and experience with cKMC. The GLM was chosen with a gamma distribution because the data were not normally distributed and exhibited skewness. Across all sub-scales, the overall model was statistically significant compared to the intercept-only model, $\chi^2$ (11) = 40.30, p < .001, indicating that the predictors collectively improved model fit. Model fit indices were satisfactory (AIC = 3568.11, BIC = 3624.12). Positive B values mean higher readiness; negative values mean lower readiness.

**Capability.** The Kruskal-Wallis test noted statistically significant differences in healthcare professionals' capability to implement cKMC across several sociodemographic and professional variables. Significant differences were observed based on residence ($\chi^2$ = 7.55, df = 1, p = .006), qualification level ($\chi^2$ = 10.70, df = 4, p = .03), age group ($\chi^2$ = 25.48, df = 3, p < .001), and work experience ($\chi^2$ = 12.15, df = 3, p = .007).

Post-hoc comparisons using Dunn's test with Bonferroni correction indicated that healthcare professionals working in urban areas were significantly more capable than those in rural areas (p = .006), with a negligible effect size ($\varepsilon^2$ = 0.01). Those with certificate-level qualifications were significantly less capable than those with a diploma (p = .008) and a bachelor's degree (p = .006), with a negligible effect size ($\varepsilon^2$ = 0.01). Age-based comparisons showed that professionals aged ≤30 years were significantly more capable than those aged 31–40 years (p < .001) and >50 years (p < .001), with a moderate effect size ($\varepsilon^2$ = 0.04). Regarding work experience, professionals with >15 years of experience perceived they were significantly more capable than those with ≤5 years (p = .02) and 6–10 years (p = .03), with a negligible effect size ($\varepsilon^2$ = 0.01) (Table 4).

**Perception.** The Kruskal-Wallis test showed statistically significant differences in perception of cKMC among healthcare professionals based on prior experience with cKMC ($\chi^2$ = 5.77, df = 1, p = .01), training history ($\chi^2$ = 6.66, df = 2, p = .03), age group ($\chi^2$ = 17.10, df = 3, p = .001), and work experience ($\chi^2$ = 19.87, df = 3, p = .001).

Post-hoc comparisons using Dunn's test with Bonferroni correction revealed that professionals with prior experience of cKMC had significantly more positive perceptions than those without (p = .01), with a negligible effect size ($\varepsilon^2$ = 0.008). Those who had received training once were significantly more positive than those who had never received training (p = .03), also with a negligible effect size ($\varepsilon^2$ = 0.008). Age-based comparisons showed that professionals aged ≤30 years had significantly more positive perceptions than those aged 31–40 years (p < .001), and 31–40 years differed significantly from 41–50 years (p = .005), with a weak effect size ($\varepsilon^2$ = 0.02). Regarding work experience, professionals with >15 years of experience were significantly more positive than those with ≤5 years (p = .001) and 6–10 years (p = .002), with a weak effect size ($\varepsilon^2$ = 0.03) (Table 4).

The GLM parameter estimates revealed that participants with prior experience on cKMC had higher perception scores (B = 0.049, SE = 0.022, 95% CI [0.005, 0.093]) compared to those without experience. The Gamma scale parameter was 0.041 (SE = 0.0025) (Table 5).

**Table 3. Descriptive characteristics of healthcare professionals on each item and subscale.**

| Factors and Items | Strongly disagree n (%) | Disagree n (%) | Agree n (%) | Strongly agree n (%) | Unsure/ uncertain n (%) | Item Mean Score (SD) | Ceiling effect (%) |
|---|---|---|---|---|---|---|---|
| **Factor 1: Capability: Subscale Median (IQR) = 31 (8) out of 9–45** | | | | | | | |
| 1. I feel competent in assisting and supporting families with cKMC practice. | 36 (6.6) | 166 (30.2) | 269 (49.0) | 61 (11.1) | 17 (3.1) | 3.28 (1.19) | 11.1 |
| 2. Assisting and supporting families with cKMC practice is part of my healthcare responsibilities. | 16 (2.9) | 66 (12) | 362 (65.9) | 97 (17.7) | 8 (1.5) | 3.83 (0.95) | 17.7 |
| 3. I am committed to promoting cKMC so that families can understand and appreciate its important benefits. | 17 (3.1) | 68 (12.4) | 365 (66.5) | 83 (15.1) | 16 (2.9) | 3.78 (0.95) | 15.1 |
| 4. I feel confident talking to families about cKMC when they have concerns about this practice. | 14 (2.6) | 112 (20.4) | 329 (59.9) | 76(13.8) | 18(3.3) | 3.62 (1.0) | 13.8 |
| 5. I provide information and promote cKMC to families in the community. | 26 (4.7) | 136 (24.8) | 297 (54.1) | 70 (12.8) | 20 (3.6) | 3.45 (1.13) | 12.8 |
| 6. My workplace provides the necessary resources and support to assist families with cKMC. | 81(14.8) | 224 (40.8) | 178 (32.4) | 39 (7.1) | 27 (4.9) | 2.76 (1.24) | 7.1 |
| 7. I encourage fathers to practice cKMC. | 42 (7.7) | 157 (28.6) | 266 (48.5) | 68 (12.4) | 16 (2.9) | 3.29 (1.22) | 12.4 |
| 8. I actively encourage and support mothers in practising cKMC as part of my healthcare responsibilities. | 13 (2.4) | 68 (12.4) | 347 (63.2) | 109 (19.9) | 12 (2.2) | 3.86 (0.95) | 19.9 |
| 9. I have received sufficient training in cKMC practice. | 140 (23.6) | 242 (40.8) | 119 (20.1) | 21(3.5) | 71 (12.0) | 2.39 (1.15) | 3.5 |
| **Factor 2: Perception: Subscale Median (IQR) = 28 (7.5) out of 8–40** | | | | | | | |
| 10. Fathers can support mothers in practising cKMC while balancing household responsibilities and childcare. | 31(5.6) | 145 (26.4) | 252 (45.9) | 103 (18.8) | 18 (3.3) | 3.46 (1.22) | 18.8 |
| 11. Family members can support mothers in practising cKMC while balancing household responsibilities and childcare. | 44 (8.0) | 134 (24.4) | 265 (48.3) | 85 (15.5) | 21 (3.8) | 3.39 (1.23) | 15.5 |
| 12. Families can feel confident handling their baby safely while practising cKMC with the proper guidance. | 11 (2.0) | 106 (19.3) | 302 (55.0) | 113 (20.6) | 17 (3.1) | 3.73 (1.05) | 20.6 |
| 13. Families who practice cKMC feel supported and accepted within their community. | 36 (6.6) | 148 (27.0) | 261(47.5) | 75 (13.7) | 29 (5.3) | 3.35 (1.19) | 13.7 |
| 14. cKMC can be integrated into family caregiving routines without disrupting daily tasks and responsibilities. | 10 (1.8) | 77 (14.0) | 345 (62.8) | 108 (19.7) | 9 (1.6) | 3.85 (0.95) | 19.7 |
| 15. Mothers feel comfortable practising cKMC in appropriate settings that ensure their privacy and modesty. | 7 (1.3) | 72 (13.1) | 340 (61.9) | 114 (20.8) | 16 (2.9) | 3.88 (0.92) | 20.8 |
| 16. Families perceive cKMC to be time-consuming.* | 13 (2.4) | 133 (24.2) | 273 (49.7) | 92 (16.8) | 38 (6.9) | 3.54 (1.10) | 16.8 |
| 17. I believe there are specific medical conditions where cKMC may not be suitable.* | 44 (8.0) | 181 (33.0) | 202 (36.8) | 65 (11.8) | 57 (10.4) | 3.11 (1.21) | 11.8 |
| **Factor 3: Knowledge: Subscale Median (IQR) = 40 (7) out of 10–50** | | | | | | | |
| 18. cKMC helps regulate the heartbeat of premature and/or low birthweight babies. | 13 (2.2) | 52 (8.8) | 314 (53.0) | 166 (28.0) | 48 (8.1) | 3.96 (0.95) | 28.0 |
| 19. KMC helps premature and low birth weight babies breathe better by providing skin-to-skin contact and reducing apnea. | 10 (1.7) | 36 (6.1) | 346 (58.3) | 164 (27.7) | 37 (6.2) | 4.04 (0.85) | 27.7 |
| 20. cKMC is a safe and effective method for caring for premature and low birth weight neonates. | 8 (1.3) | 27(4.6) | 318 (53.6) | 206 (34.7) | 34 (5.7) | 4.16 (0.82) | 34.7 |
| 21. cKMC is recommended for a minimum of eight hours per day to promote the health of premature and/or low birth weight neonates. | 10 (1.7) | 79 (13.3) | 258 (43.5) | 129 (21.8) | 117 (19.7) | 3.7 (1.00) | 21.8 |

*(Continued)*

**Table 3.** (Continued)

| Factors and Items | Strongly disagree n (%) | Disagree n (%) | Agree n (%) | Strongly agree n (%) | Unsure/ uncertain n (%) | Item Mean Score (SD) | Ceiling effect (%) |
|---|---|---|---|---|---|---|---|
| 22. I am aware of KMC practice, which comprises skin-to-skin contact and exclusive breastfeeding for premature and/or low birth weight neonates. | 29 (4.9) | 91(15.3) | 289 (48.7) | 145 (24.5) | 39 (6.6) | 3.73 (1.13) | 24.5 |
| 23. cKMC helps mothers successfully breastfeed premature and low birth-weight neonates. | 9 (1.5) | 29 (4.9) | 337(56.8) | 200 (33.7) | 18 (3.0) | 4.16 (0.82) | 33.7 |
| 24. I am aware of specially designed baby wraps that help parents provide continuous skin-to-skin care safely for their newborns. | 11(1.9) | 131(22.1) | 269 (45.4) | 130 (21.9) | 52 (8.8) | 3.63 (1.10) | 21.9 |
| 25. KMC can be practised at home or in the community for premature and low birth weight neonates. | 14 (2.4) | 49 (8.3) | 343(57.8) | 155 (26.1) | 32 (5.4) | 3.97 (0.92) | 26.1 |
| 26. cKMC can help increase a mother's milk supply for her newborn. | 7 (1.2) | 48 (8.1) | 306 (51.6) | 176 (29.7) | 56 (9.4) | 4.01 (0.90) | 29.7 |
| 27. Neonates may experience slight temperature fluctuation during cKMC.* | 9 (1.5) | 102 (17.2) | 278 (46.8 | 154 (26.0) | 50 (8.4) | 3.79 (1.06) | 26.0 |
| **Factor 4: Engagement: Subscale Median (IQR) = 12 (4) out of 3–15** | | | | | | | |
| 28. Strong collaboration between health facilities and the community can enhance cKMC implementation. | 5 (0.9) | 25 (4.6) | 342 (62.3) | 175 (31.9) | 2 (0.4) | 4.20 (0.74) | 31.9 |
| 29. Increased financial support from governments and non-governmental organisations can further strengthen cKMC training for healthcare providers. | 8 (1.5) | 22 (4.0) | 306 (55.7) | 206 (37.5) | 7 (1.3) | 4.24 (0.79) | 37.5 |
| 30. With appropriate resources and support, cKMC can be effectively practised in remote areas | 6 (0.1) | 37 (6.7) | 358 (65.2) | 134 (24.4) | 14 (2.6) | 4.05 (0.79) | 24.4 |

* This indicates that the scores for certain items have been reversed because they were originally worded in a way that resulted in negative responses.

**Knowledge.** Significant differences in knowledge of cKMC were found across qualification levels ($\chi^2 = 12.40$, df = 4, $p = .015$), age groups ($\chi^2 = 64.27$, df = 4, $p < .001$), and work experience ($\chi^2 = 20.15$, df = 3, $p < .001$).

Post-hoc analysis indicated a significant difference in knowledge between professionals with certificate-level qualifications and those with a diploma ($p = .002$) and a bachelor's degree ($p = .02$), with a negligible effect size ($\varepsilon^2 = 0.01$). Age-based comparisons showed that professionals aged ≤ 30 years had significantly higher knowledge than those aged 31–40 years ($p < .001$), 41–50 years ($p = .003$), and >50 years ($p < .001$), with a moderate effect size ($\varepsilon^2 = 0.1$). For work experience, those with 6–10 years differed significantly from those with >15 years ($p = .002$), and from those with 11–15 years ($p = .04$), with a weak effect size ($\varepsilon^2 = 0.02$) (Table 4).

The GLM parameter estimates demonstrated that knowledge scores increased with age. Specifically, participants under 21 years ($B = -1.028$, $SE = 0.105$, 95% CI [–1.234, –0.822], $p < .001$), aged 21–30 years ($B = -0.132$, $SE = 0.028$, $p < .001$), and aged 31–40 years ($B = -0.052$, $SE = 0.024$, $p = .027$) had significantly lower knowledge compared to those aged over 50 years. The Gamma scale parameter was 0.020 ($SE = 0.0012$) (Table 5).

**Engagement.** The Kruskal-Wallis test indicated significant differences in engagement with cKMC based on residence ($\chi^2 = 9.46$, df = 1, $p = .002$), qualification level ($\chi^2 = 54.04$, df = 4, $p = .001$), prior experience with cKMC ($\chi^2 = 8.11$, df = 1, $p = .004$), training history ($\chi^2 = 9.34$, df = 2, $p = .009$), age group ($\chi^2 = 228.75$, df = 3, $p < .001$), and work experience ($\chi^2 = 202.22$, df = 3, $p < .001$).

Post-hoc comparisons indicated that professionals in urban areas were significantly more engaged than those in rural areas (p = .002), with a negligible effect size ($\varepsilon^2 = 0.01$). Certificate-level qualification holders were significantly less engaged than diploma ($p < .001$) and bachelor's degree holders ($p = .013$), showing a moderate effect size ($\varepsilon^2 = 0.09$).

**Table 4. Comparing healthcare professionals' readiness of cKMC with sociodemographic and professional characteristics using the Kruskal-Wallis test, post hoc and effect sizes.**

| Components | Comparison variable | Comparison groups | Median and Interquartile Range | Test Statistics | Degree of freedom | p-value | Post-hoc comparison using Dunn's test with Bonferroni correction | | Effect size (Epsilon Squared (ε²)) |
|---|---|---|---|---|---|---|---|---|---|
| | | | | | | | Group to group | Adjusted p-value | |
| Capability | Residence | Urban | 32 (9) | 7.55 | 1 | 0.006 | | | 0.01 |
| | | Rural | 30 (8) | | | | | | |
| | Qualifications | Certificate | 30 (9) | 10.70 | 4 | 0.03 | Certificate- diploma | 0.008 | 0.01 |
| | | Diploma | 31 (9) | | | | Certificate – Bachelor | 0.006 | |
| | | Bachelor | 31 (8) | | | | | | |
| | | Master | 33 (13) | | | | | | |
| | Age | ≤30 years | 30 (9) | 25.48 | 3 | <0.001 | ≤30 yrs −31–40 yrs | <0.001 | 0.04 |
| | | 31-40 years | 32 (8) | | | | ≤30 yrs ->50 yrs | <0.001 | |
| | | 41-50 years | 30.5 (8) | | | | | | |
| | | >50 years | 32 (9) | | | | | | |
| | Work Experience | ≤5 years | 30 (9) | 12.15 | 3 | 0.007 | ≤ 5 yrs −11–15 yrs | 0.02 | 0.01 |
| | | 6-10 years | 30.50 (9) | | | | 6-10 yrs->15 yrs | 0.03 | |
| | | 11-15 years | 31 (9) | | | | | | |
| | | >15 years | 31 (8) | | | | | | |
| Perception | Experience of cKMC | Yes | 29 (8) | 5.77 | 1 | 0.01 | | | 0.008 |
| | | No | 28 (8) | | | | | | |
| | Training | Never | 28 (8) | 6.66 | 2 | 0.03 | Never-One time | 0.03 | 0.008 |
| | | One time | 30 (9) | | | | | | |
| | | Two and more | 28 (7) | | | | | | |
| | Age | ≤30 years | 30 (9) | 17.10 | 3 | 0.001 | ≤30 yrs −31–40 yrs | <0.001 | 0.02 |
| | | 31-40 years | 28 (9) | | | | 31-40 yrs −41–50 yrs | 0.005 | |
| | | 41-50 years | 32 (9) | | | | | | |
| | | >50 years | 29 (10) | | | | | | |
| | Work Experience | ≤5 years | 27.5 (5) | 19.87 | 3 | 0.001 | ≤5 yrs ->15 yrs | 0.001 | 0.03 |
| | | 6-10 years | 28 (8) | | | | 6-10 yrs>15 yrs | 0.002 | |
| | | 11-15 years | 27 (9) | | | | | | |
| | | >15 years | 30 (8) | | | | | | |
| Engagement | Residence | Urban | 12 (2) | 9.46 | 1 | 0.002 | | | 0.01 |
| | | Rural | 12 (1) | | | | | | |
| | Qualifications | Certificate | 12 (0) | 54.04 | 4 | 0.001 | Certificate- diploma | <0.001 | 0.09 |
| | | Diploma | 13 (2) | | | | Certificate – Bachelor | 0.013 | |
| | | Bachelor | 12 (1) | | | | | | |
| | | Master | 12 (1) | | | | | | |
| | Experience working with KMC in a health facility | Yes | 12 (2) | 8.11 | 1 | 0.004 | | | 0.01 |
| | | No | 12 (1) | | | | | | |
| | Training | Never | 12 (1) | 9.34 | 2 | 0.009 | Never – one time | 0.007 | 0.01 |
| | | One time | 13 (3) | | | | | | |
| | | Two and more | 13 (1) | | | | | | |

*(Continued)*

**Table 4.** (Continued)

| Compo-nents | Comparison variable | Comparison groups | Median and Interquartile Range | Test Statistics | Degree of freedom | p-value | Post-hoc comparison using Dunn's test with Bonferroni correction | | Effect size (Epsilon Squared (ε²)) |
|---|---|---|---|---|---|---|---|---|---|
| | | | | | | | Group to group | Adjusted p-value | |
| | Age | ≤30 years | 12 (1) | 228.75 | 3 | <0.001 | ≤30 yrs – 31–40 yrs | <0.001 | 0.41 |
| | | 31-40 years | 12 (1) | | | | ≤30 yrs – 41–50 yrs | <0.001 | |
| | | 41-50 years | 14 (2) | | | | ≤30 yrs –>50 yrs | <0.001 | |
| | | >50 years | 14 (2) | | | | | | |
| | Work Experience | ≤5 years | 12 (2) | 202.22 | 3 | <0.001 | ≤5 yrs −11–15 yrs | <0.001 | 0.36 |
| | | 6-10 years | 12 (0) | | | | ≤5 yrs -> 15 yrs | <0.001 | |
| | | 11-15 years | 12 (1) | | | | 6-10 yrs > 15 yrs | <0.001 | |
| | | >15 years | 14 (2) | | | | ≤5 yrs −11–15 yrs | <0.001 | |
| Knowl-edge | Qualifications | Certificate | 38 (9) | 12.40 | 4 | 0.015 | Certificate- diploma | 0.002 | 0.01 |
| | | Diploma | 40 (7) | | | | Certificate – Bachelor | 0.02 | |
| | | Bachelor | 40 (7) | | | | | | |
| | | Master | 36 (9) | | | | | | |
| | Age | ≤30 years | 38 (8) | 64.27 | 4 | <0.001 | ≤30 yrs −31–40 yrs | <0.001 | 0.1 |
| | | 31-40 years | 40 (7) | | | | ≤30 yrs −41–50 yrs | 0.003 | |
| | | 41-50 years | 40.5 (7) | | | | ≤30 yrs -> 50 yrs | <0.001 | |
| | | >50 years | 42.5 (7) | | | | | | |
| | Work Experience | ≤5 years | 39 (6) | 20.15 | 3 | <0.001 | 6-10 yrs-> 15 yrs | 0.002 | 0.02 |
| | | 6-10 years | 39 (7) | | | | 6-10 yrs – 11–15 | 0.04 | |
| | | 11-15 years | 41.5 (7) | | | | | | |
| | | >15 years | 40 (6) | | | | | | |

**Table 5. Predictors of dimensions of healthcare professionals' readiness to implement cKMC.**

| Dependent Variable | Significant Predictors | B (SE) | 95% CI | Wald χ² | p |
|---|---|---|---|---|---|
| Capability | Age (31–40) years | −0.004 (.037) | [−0.077, 0.069] | 0.013 | .910 |
| | Age (41–50) years | −0.069 (.036) | [−0.139, 0.001] | 3.699 | .054 |
| | Reference >50 years | | | | |
| Perception | Experience on cKMC (Yes) | 0.049 (.022) | [0.005, 0.093] | 4.781 | .029* |
| | Reference (No) | | | | |
| Knowledge | Age <21 | −1.028 (.105) | [−1.234, −0.822] | 95.52 | <.001* |
| | Age 21–30 years | −0.132 (.028) | [−0.187, −0.078] | 22.86 | <.001* |
| | Age 31–40 years | −0.052 (.024) | [−0.099, −0.006] | 4.91 | .027* |
| | Reference >50 years | | | | |
| Engagement | Age 21–30 | −0.130 (.028) | [−0.185, −0.075] | 21.34 | <.001* |
| | Reference >50 years | | | | |
| | Work exp. <5 yrs | −0.115 (.026) | [−0.165, −0.065] | 20.18 | <.001* |
| | Reference >15 years | | | | |

* Indicates significant association

Professionals with prior cKMC experience were more engaged than those without (p = .004), and those trained once were more engaged than those never trained (p = .007), both with negligible effect sizes (ε² = 0.01). Age comparisons revealed that professionals aged ≤ 30 years had significantly higher engagement than older groups (p < .001), with a strong effect size (ε² = 0.41). Additionally, those with >15 years of experience showed significantly higher engagement than those with ≤5 years, 6–10 years, and 11–15 years (p < .001), also with a strong effect size (ε² = 0.36).) (Table 4).

As the GLM parameter estimates revealed that participants aged 21–30 years (B = –0.130, SE = 0.028, 95% CI [–0.185, –0.075], p < .001) and those with fewer than five years of work experience (B = –0.115, SE = 0.026, 95% CI [–0.165, –0.065], p < .001) demonstrated lower engagement levels compared to their respective reference groups. The Gamma scale parameter was 0.018 (SE = 0.0011) (Table 5).

## Discussion

This study developed and validated a new instrument to measure healthcare professionals' readiness to implement cKMC. Findings demonstrated strong reliability, strong content validity, and four factors of readiness, but exposed persistent system-level barriers.

The findings highlighted a reassuring level of awareness and positive attitudes toward cKMC, while also revealing significant gaps in training, institutional support, and practical capability. These insights are consistent with previous research emphasising the critical importance of healthcare professionals' readiness for the successful scaling-up of cKMC initiatives in low- and middle-income countries [12,16].

The healthcare professionals perceived a strong foundational understanding of the physiological and clinical advantages of cKMC. High agreement on items related to heartbeat regulation, respiratory support, and breastfeeding benefits reflects alignment with established evidence on the efficacy of skin-to-skin contact for premature and low birthweight neonates.

However, item-level analysis reveals notable gaps in applied knowledge. Low awareness of baby wraps and the feasibility of home-based care suggests limited familiarity with practical tools and community-level implementation strategies. These findings suggest a critical distinction between conceptual awareness and practical readiness. Knowledge alone does not ensure implementation; it must be contextualised within clinical workflows, cultural norms, and community realities. As previous studies have demonstrated, effective adoption of cKMC requires experiential learning, hands-on training, and community-based demonstrations that reinforce practical competencies [48,49].

Furthermore, the utilisation of guidelines to support healthcare professionals is particularly low, at just 13.2%. These observations are consistent with previous studies in Ethiopia, which have indicated a partial understanding of KMC concepts and guidelines among healthcare professionals [19,50,51]. The comparatively higher knowledge levels observed among younger healthcare providers and those with bachelor's degrees may be linked to recent updates in pre-service curricula that include concepts of the evidence to improve utilisation of the outcome [52,53]. In contrast, older or more experienced professionals may not have benefited from such structured training. This generational gap suggests the necessity for ongoing professional development, standardised refresher courses, and effective strategies for disseminating evidence, guidelines, and protocols that are essential for successful implementation in both community and home settings.

This study identified significant gaps in professional confidence, commitment, and autonomy in the implementation of cKMC. Twenty-three per cent of participants reported low confidence in promoting cKMC, especially in environments lacking resources or leadership buy-in. Confidence and commitment are not fixed personal traits but are shaped by external factors such as the work environment, institutional expectations, and the clarity of role definitions [54,55]. When these elements are well-aligned, they contribute to the cultivation of a robust sense of assurance, enhancing their capacity to deliver cKMC effectively [55].

This study attests that over half of the healthcare professionals reported that their workplace lacked resources and institutional support to facilitate cKMC, paralleling findings from studies in Malawi and Uganda that identified a shortage

of training, infrastructure, workload, and logistical barriers as central impediments to KMC scale-up [56,57]. Urban professionals perceived that they were more capable than their rural counterparts, which may reflect greater exposure to facility-based KMC, stronger referral linkages, and closer supervision from district health offices.

In building healthcare professionals' confidence in practice, structured training, mentorship, and policy integration are essential [48,58]. When cKMC is embedded in governmental policies, healthcare professionals are more likely to feel empowered and accountable to implement KMC in the community and home settings.

This study suggests that healthcare professionals believe that families can play a supportive role in adopting cKMC, especially when provided with appropriate guidance and culturally sensitive environments. The highest-rated item was maternal comfort in practising cKMC in settings that ensure privacy and modesty, reinforcing the importance of dignity and cultural sensitivity in care environments [59,60]. Similarly, the integration of cKMC into daily routines without disrupting household responsibilities was viewed positively, suggesting that cKMC is seen as feasible and sustainable with adequate support.

In contrast, lower mean scores for items related to certain medical conditions indicate that cKMC may not be deemed suitable in those cases. This suggests a division or uncertainty among healthcare professionals regarding the clinical boundaries of cKMC. Additionally, the participants' attitudes related to family involvement suggests limited shared caregiving, potentially influenced by embedded gender norms and a lack of targeted interventions to engage broader family networks [61,62].

The study observed broadly positive perceptions of cKMC, although about one-third of participants expressed doubts regarding community and family support. Those with prior cKMC experience exhibited significantly more favourable perceptions, consistent with the experiential learning hypothesis where knowledge is created through the transformation of experience [63]. However, concerns about time investment required and the cultural acceptability of skin-to-skin contact remain salient in Ethiopian communities, as also reported in India and Ghana contexts [64,65]. These findings suggest that sociocultural and gender dynamics, such as modesty, household burden, and paternal involvement, continue to shape health workers' perceptions and counselling practices.

Healthcare professionals highlighted the importance of collaboration, financial support, and resource availability. System-level engagement is widely recognised in literature as critical for sustainable cKMC implementation. The literature consistently shows that cKMC thrives in environments with strong leadership, dedicated space, and integrated service delivery [66]. A systematic review on KMC implementation recommended regular training and national policy enforcement to ensure consistency across facilities and communities [48]. Without institutional commitment, even the most knowledgeable and confident healthcare professionals are limited in their ability to implement cKMC. Alignment between policy frameworks, financing, and infrastructure is needed to support frontline efforts in the implementation of cKMC in community and home settings.

Engagement levels were high in principle but limited in structured community practice. Only half of the respondents reported conducting home visits or community awareness activities. Engagement was significantly associated with age, experience, and urban residence, with younger and less experienced professionals showing lower involvement. This result partially contrasts with findings from India and Bangladesh, where community health workers played a central role in sustaining cKMC through household follow-up [16,67]. The discrepancy may be due to Ethiopia's fragmented community health infrastructure following regional instability, as well as the absence of designated cKMC focal persons at the primary-care level.

Age emerged as a consistent predictor for knowledge and engagement, while prior cKMC experience predicted perception. Work experience also significantly influenced engagement, suggesting that practical exposure rather than academic qualification is the key determinant of readiness. Formal training did not significantly predict any readiness dimension, suggesting that existing training programs may be too brief or theoretical to translate into competence, a concern that resonates with similar evaluations in Indonesia [68,69].

The current findings show a more fragmented readiness pattern, emphasising the contextual challenges of decentralising neonatal care. Some implementation research reported rapid adoption with strong supervisory and policy support [67,70]. This study reveals that Ethiopia's health workforce remains underprepared for the community transition of cKMC. Challenges likely stem from systemic variations, particularly in financing, mentorship, and digital integration. Strengthening cKMC readiness requires integrated training, consistent policy reinforcement, and digital mentorship strategies to sustain neonatal care in resource-limited settings.

### Implications for practice and policy

The HiPECK-cKMC instrument provides a practical framework to identify barriers and guide the development of interventions on cKMC implementation. While healthcare professionals recognise the value of cKMC, deficits in practical skills and system support remain. Integrating cKMC into preservice and in-service curricula, along with case-based and simulation training, can strengthen readiness in low-resource contexts.

At the system level, institutional and policy support are vital for scaling up. Investment in infrastructure, supervision, and performance monitoring will enhance sustainability. Embedding cKMC indicators into Ethiopia's Health Extension Program and maternal–child health systems may ensure accountability. Family-centred, culturally sensitive approaches that engage fathers and extended families are essential for community acceptance.

The HiPECK-cKMC can also serve as a readiness assessment and monitoring tool, guiding resource allocation and measuring progress. Broader collaboration among government, NGOs, and communities is required to integrate cKMC into existing health structures and ensure continuity of care. While the instrument was developed for the Ethiopian context, the tool has potential relevance for other low-resource settings; however, cultural adaptation and context-specific validation are required before wider application.

### Limitations

The instrument was tested in a single low-resource healthcare context. Self-reported responses may introduce biases, such as social desirability and recall bias, leading participants to overstate their knowledge or commitment, especially when questions were asked verbally by a research assistant.

While data was collected from many participants, some information was missing due to the branching logic in the design, excluding those unaware of cKMC from questions about attitudes, practices, and barriers. Engagement has emerged as a significant factor; however, the subscale consists of only three items and recorded the lowest Cronbach's Alpha value of 0.67. This may not adequately capture the intricacies of systemic enablers and barriers. As a result, there is a need for expansion to encompass aspects such as leadership, policy integration, and resource allocation.

The lower internal consistency observed in some engagement items highlights the necessity for further refinement. Future efforts will focus on revising or expanding these items to better reflect the multidimensional nature of engagement with cKMC, followed by additional psychometric testing with larger sample sizes. Further testing, including confirmatory factor analysis, multi-site samples, and test-retest analysis, is required.

### Conclusion

The HiPECK-cKMC instrument offers a validated framework for assessing the readiness of healthcare professionals to implement cKMC. The scale captures four interrelated domains, perception, engagement, capability, and knowledge, which offer a structured framework for understanding the multifaceted nature of cKMC implementation. It may support the identification of readiness levels, guide targeted interventions, inform policy development, and monitor progress in implementation.

This study provides evidence on healthcare professionals' readiness to implement cKMC within a resource-limited context in Ethiopia. Although foundational awareness is strong, readiness gaps persist, particularly in training, system

support, and engagement. These insights emphasise the necessity for integrated strategies that reinforce both individual competencies and institutional capacity. Factors influencing healthcare professionals' readiness to implement cKMC primarily include age, experience, and prior exposure to cKMC, rather than academic qualifications or the frequency of training. The findings highlight the critical need to translate awareness into practice through structured mentorship, supportive supervision, and targeted policy interventions. Furthermore, enhancing digital and community platforms for knowledge sharing could significantly contribute to the sustainability of these practices.

National cKMC scale-up in Ethiopia requires integrated workforce, community, and system reform, supported by continued validation, cross-cultural and longitudinal research to assess program effectiveness, sustain adoption, and inform evidence-based neonatal policy.

## Supporting information

**S1 File. Checklist.**
(DOCX)

**S2 File. EFA Dataset.**
(SAV)

**S3 File. HiPECK-cKMC Instrument.**
(DOCX)

## Acknowledgments

We thank the study participants for their time and contribution to this research. We also thank the experts who reviewed the instrument and contributed to the content validity evaluation, and the research assistants who supported data collection. We acknowledge the contributions of team members who assisted with language translation of the instrument but did not meet the criteria for authorship.

## Author contributions

**Conceptualization:** Hagos T. Atalay, Victoria J. Kain, Amanda G. Carter.

**Data curation:** Hagos T. Atalay, Victoria J. Kain, Amanda G. Carter.

**Formal analysis:** Hagos T. Atalay, Victoria J. Kain, Amanda G. Carter.

**Funding acquisition:** Hagos T. Atalay, Victoria J. Kain, Amanda G. Carter.

**Investigation:** Hagos T. Atalay, Victoria J. Kain, Amanda G. Carter.

**Methodology:** Hagos T. Atalay, Victoria J. Kain, Amanda G. Carter.

**Project administration:** Hagos T. Atalay, Victoria J. Kain, Amanda G. Carter.

**Resources:** Hagos T. Atalay, Victoria J. Kain, Amanda G. Carter.

**Software:** Hagos T. Atalay, Victoria J. Kain, Amanda G. Carter.

**Supervision:** Victoria J. Kain, Amanda G. Carter.

**Validation:** Hagos T. Atalay, Victoria J. Kain, Amanda G. Carter.

**Visualization:** Hagos T. Atalay, Victoria J. Kain, Amanda G. Carter.

**Writing – original draft:** Hagos T. Atalay.

**Writing – review & editing:** Hagos T. Atalay, Victoria J. Kain, Amanda G. Carter.

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
