## [Decision Letter · Decision Letter 0]

2 Jan 2026

PONE-D-25-57353Evaluating healthcare professionals’ readiness for community-based kangaroo mother care in resource-limited setting

PLOS One

Dear Dr. Atalay,

Thank you for submitting your manuscript to PLOS ONE. After careful consideration, we feel that it has merit but does not fully meet PLOS ONE’s publication criteria as it currently stands. Therefore, we invite you to submit a revised version of the manuscript that addresses the points raised during the review process.

Please revise your manuscript to clearly and explicitly present your study objectives and research questions. Clearly stating these elements will help readers understand the purpose and scope of your research. Additionally, we encourage you to carefully review the presentation of your results to ensure clarity, logical flow, and alignment with your objectives. Clear tables, figures, and narrative descriptions can significantly enhance the readability and impact of your findings.<o:p></o:p>

We look forward to receiving your revised manuscript.

Kind regards,

Addis Eyeberu

Academic Editor

PLOS One

Journal Requirements:

Additional Editor Comments:

Dear Authors,

Please revise your manuscript to clearly and explicitly present your study objectives and research questions. Clearly stating these elements will help readers understand the purpose and scope of your research. Additionally, we encourage you to carefully review the presentation of your results to ensure clarity, logical flow, and alignment with your objectives. Clear tables, figures, and narrative descriptions can significantly enhance the readability and impact of your findings.

Reviewers' comments:

Reviewer's Responses to Questions

Comments to the Author

1. Is the manuscript technically sound, and do the data support the conclusions?

Reviewer #1: Yes

Reviewer #2: Yes

Reviewer #3: Yes

2. Has the statistical analysis been performed appropriately and rigorously? 

Reviewer #1: No

Reviewer #2: Yes

Reviewer #3: Yes

3. Have the authors made all data underlying the findings in their manuscript fully available?

Reviewer #1: No

Reviewer #2: Yes

Reviewer #3: No

4. Is the manuscript presented in an intelligible fashion and written in standard English?

Reviewer #1: Yes

Reviewer #2: Yes

Reviewer #3: Yes

5. Review Comments to the Author

Reviewer #1: The manuscript is interesting and valuable. However, there are some concerns which should be addressed before its acceptance. Following are some of the concerns.

• Please explain the term “certificate” in the context of the qualification.

• The Results section can be summarized more succinctly. Much of its content duplicates what is already reported in the tables.

• When reporting attainable scores, provide both the minimum and maximum scores. Reporting only the maximum can be misleading; for example, in line 298, 45 should be reported as 9–45.

Table 1

• Please include the number of participants at the end of the table title.

• Measurement units should be mentioned for the variables (e.g., age, work experience) after the variable name.

• For dichotomous variables, reporting one category is sufficient and will shorten the table.

• Consider abbreviating and shortening the names of variables.

Table 3

• There appear to be missing values in some items. For example, the sum of responders for item 1 is 543 and for item 2 is 549. Please add a column reporting the number of responders for each item and explain how missing values were handled.

• Some items, including item 2, may not be suited for the reported domain (capability). Item 9 does not appear to fit with the response format.

• Please explain how each item was scored and indicate which items are reverse-scored. State the attainable (possible) score for each domain.

General statistical reporting

• You state that a Kruskal–Wallis test was used for between-group comparisons due to skewness. In Table 3, however, you report mean (SD). When skewness is present, mean (SD) is not the best descriptive summary; consider reporting the median with interquartile range (Q1–Q3) instead.

Table 4

• Descriptive results should also be reported. For example, although the table shows a significant difference between participants from urban and rural areas in terms of capability score, it does not indicate which group has the higher score.

Table 5

• The reference group should be specified for each predictor.

Additional items

• Report ceiling effects where relevant.

• Stability is also recommended as an indicator of reliability. If you have not assessed it, acknowledge this in the limitations subsection.

Reviewer #2: This manuscript addresses an important gap in neonatal and community-based care by developing and preliminarily validating a tool to assess healthcare professionals’ readiness to implement community-based kangaroo mother care (cKMC) in low-resource settings. The study is timely, methodologically sound, and highly relevant to health systems strengthening in LMICs. The large sample size, clear theoretical grounding, and comprehensive psychometric evaluation strengthen the contribution of this work.

Strengths:

- The use of established frameworks (DeVellis and the TDF) provides strong conceptual grounding.

- The sample size is adequate for factor analysis and reliability testing.

- Ethical approvals and data availability are clearly described.

- The instrument demonstrates acceptable to good internal consistency and construct validity.

Minor points for improvement:

1. The manuscript would benefit from a brief clarification on why PCA rather than common factor analysis was selected, and how this choice may affect interpretation.

2. Some items in the Engagement subscale show lower internal consistency; a short discussion on plans for future refinement or item expansion would strengthen the paper.

3. Minor grammatical and stylistic edits are needed throughout the manuscript to improve readability.

4. The discussion could be strengthened by briefly addressing how the tool may be adapted or validated in other low-resource or non-Ethiopian contexts.

Conclusion:

Overall, this is a well-conducted and clearly reported study. I recommend acceptance after minor revisions addressing the points above.

Reviewer #3: General Assessment

This manuscript presents the development and preliminary testing of an instrument to assess healthcare professionals’ readiness for community-based kangaroo mother care (cKMC). The topic is important, and the large sample size is a strength. However, clarification of the study’s primary objective and methodological refinement are required.

Major Comments:

1. The manuscript needs to clearly state whether the primary objective is to develop and preliminarily validate an instrument or to evaluate healthcare professionals’ readiness.

The authors should clearly distinguish between instrument development as the primary objective and exploratory description of readiness as a secondary output. Readiness-related findings should be framed cautiously and not as confirmatory conclusions.

The manuscript needs to clearly state whether the primary objective is to develop and preliminarily validate an instrument or to evaluate healthcare professionals’ readiness.

2. . Conceptual Framework

The manuscript indicates that 14 domains from the Theoretical Domains Framework were grouped into four constructs (capability, perception, knowledge, engagement). The rationale for this reduction is insufficiently explained. A clear explanation or a mapping table linking the original domains to the final constructs would improve conceptual clarity.

3. Interpretation of Statistical Findings

Several associations reach statistical significance but are accompanied by very small effect sizes, indicating limited practical relevance. The discussion should therefore differentiate more clearly between statistical significance and real-world importance.

Minor comment:

4. Terminology related to cKMC is used inconsistently (e.g., “community-based KMC,” “cKMC,” “home-based KMC”). Consistent use of a single term throughout the manuscript would improve clarity.

5. Consent process should be stated earlier in the Methods

Example wording: Information regarding informed consent (e.g., that questionnaire completion implied consent) is mentioned later in the manuscript. For clarity, this information should be stated earlier in the Methods section.

6. PLOS authors have the option to publish the peer review history of their article (what does this mean?). If published, this will include your full peer review and any attached files.

Do you want your identity to be public for this peer review? For information about this choice, including consent withdrawal, please see our Privacy Policy.

Reviewer #1:  Yes: Prof. Sakineh Mohammad-Alizadeh-Charandabi

Reviewer #2:  Yes: Chinedum Favour Ajala

Reviewer #3: No

---

## [Author Response · Author response to Decision Letter 1]

29 Jan 2026

Dear Reviewers and Editor,

We greatly appreciate your valuable feedback and suggestions. We have taken the time to address all of your comments comprehensively. A detailed point-by-point response to each of your observations has been included in the submission system for your review.

---

## [Decision Letter · Decision Letter 1]

14 Apr 2026

PONE-D-25-57353R1Evaluating healthcare professionals’ readiness for community-based kangaroo mother care in resource-limited settingPLOS One

Dear Dr. Atalay,

Thank you for submitting your manuscript to PLOS ONE. After careful consideration, we feel that it has merit but does not fully meet PLOS ONE’s publication criteria as it currently stands. Therefore, we invite you to submit a revised version of the manuscript that addresses the points raised during the review process.<section class="text-token-text-primary w-full focus:outline-none [--shadow-height:45px] has-data-writing-block:pointer-events-none has-data-writing-block:-mt-(--shadow-height) has-data-writing-block:pt-(--shadow-height) [&:has([data-writing-block])>*]:pointer-events-auto [content-visibility:auto] supports-[content-visibility:auto]:[contain-intrinsic-size:auto_100lvh] R6Vx5W_threadScrollVars scroll-mb-[calc(var(--scroll-root-safe-area-inset-bottom,0px)+var(--thread-response-height))] scroll-mt-[calc(var(--header-height)+min(200px,max(70px,20svh)))]" data-scroll-anchor="false" data-testid="conversation-turn-2" data-turn="assistant" data-turn-id="request-WEB:e3d9f303-a3dc-43f5-a4e1-b61693ccdf3a-0" dir="auto">

Please thoroughly revise the entire paper to correct typographical errors and formatting issues, ensuring full alignment with PLOS ONE requirements.

</section>Please submit your revised manuscript by May 29 2026 11:59PM. If you will need more time than this to complete your revisions, please reply to this message or contact the journal office at plosone@plos.org. Please include the following items when submitting your revised manuscript:

We look forward to receiving your revised manuscript.

Kind regards,

Addis Eyeberu

Academic Editor

PLOS One

Journal Requirements:

Reviewers' comments:

Reviewer's Responses to Questions

Comments to the Author

1. If the authors have adequately addressed your comments raised in a previous round of review and you feel that this manuscript is now acceptable for publication, you may indicate that here to bypass the “Comments to the Author” section, enter your conflict of interest statement in the “Confidential to Editor” section, and submit your "Accept" recommendation.

Reviewer #3: All comments have been addressed

Reviewer #4: (No Response)

2. Is the manuscript technically sound, and do the data support the conclusions?

Reviewer #3: Yes

Reviewer #4: (No Response)

3. Has the statistical analysis been performed appropriately and rigorously? 

Reviewer #3: Yes

Reviewer #4: (No Response)

4. Have the authors made all data underlying the findings in their manuscript fully available?

Reviewer #3: Yes

Reviewer #4: Yes

5. Is the manuscript presented in an intelligible fashion and written in standard English?

Reviewer #3: Yes

Reviewer #4: Yes

6. Review Comments to the Author

Reviewer #3: The manuscript describes a technically sound piece of scientific research with data that support the conclusions. The study followed the CROSS and DeVellis’s guidelines, employing a robust multi-stage process for the development and validation of the HiPECK-cKMC instrument.

Technical Rigor and Statistical Analysis

The methodology is rigorous, utilizing a sample size well-powered for psychometric testing. The authors demonstrated high statistical standards by confirming non-normal data distribution and appropriately applying non-parametric tests, such as the Kruskal-Wallis test and Dunn’s test with Bonferroni correction. Furthermore, the use of Generalised Linear Models with a Gamma distribution effectively modeled the skewed survey data. The study established strong construct validity through Exploratory Factor Analysis and demonstrated high internal consistency.

Data and Conclusions

The findings regarding significant gaps in formal training and guideline familiarity directly support the conclusion that, despite positive perceptions, there is a critical need for targeted training and policy support. The authors also correctly addressed previous concerns regarding effect sizes and terminology consistency, ensuring that the discussion differentiates between statistical significance and practical relevance.

Ethics

The study adheres to ethical standards, with approvals from relevant boards in Ethiopia and Australia and a clear statement indicating that survey completion constituted implied informed consent. The research provides a validated tool and actionable insights suitable for publication.

Reviewer #4: (No Response)

7. PLOS authors have the option to publish the peer review history of their article (what does this mean?). If published, this will include your full peer review and any attached files.

Do you want your identity to be public for this peer review? For information about this choice, including consent withdrawal, please see our Privacy Policy.

Reviewer #3: No

Reviewer #4: No

---

## [Author Response · Author response to Decision Letter 2]

23 Apr 2026

We addressed all concerns raised by reviewer. Please refer to the attached response table in the submission system.

---

## [Editor Report · Decision Letter 2]

26 Apr 2026

Evaluating healthcare professionals’ readiness for community-based kangaroo mother care in resource-limited setting

PONE-D-25-57353R2

Dear Dr. Atalay,

We’re pleased to inform you that your manuscript has been judged scientifically suitable for publication and will be formally accepted for publication once it meets all outstanding technical requirements.

Kind regards,

Addis Eyeberu

Academic Editor

PLOS One

Additional Editor Comments (optional):

All comments have been addressed thoroughly and effectively.
---

## [Editor Report · Acceptance letter]

PONE-D-25-57353R2

PLOS One

Dear Dr. Atalay,

I'm pleased to inform you that your manuscript has been deemed suitable for publication in PLOS One. Congratulations! Your manuscript is now being handed over to our production team.

Kind regards,

on behalf of

Dr. Addis Eyeberu

Academic Editor

PLOS One